# Multispecies biofilm architecture determines bacterial exposure to phages

**James B. Winans, Benjamin R. Wucher, Carey D. Nadell** *

Department of Biological Sciences, Dartmouth, Hanover, New Hampshire, United States of America

* carey.d.nadell@dartmouth.edu

## Abstract

Numerous ecological interactions among microbes—for example, competition for space and resources, or interaction among phages and their bacterial hosts—are likely to occur simultaneously in multispecies biofilm communities. While biofilms formed by just a single species occur, multispecies biofilms are thought to be more typical of microbial communities in the natural environment. Previous work has shown that multispecies biofilms can increase, decrease, or have no measurable impact on phage exposure of a host bacterium living alongside another species that the phages cannot target. The reasons underlying this variability are not well understood, and how phage–host encounters change within multispecies biofilms remains mostly unexplored at the cellular spatial scale. Here, we study how the cellular scale architecture of model 2-species biofilms impacts cell–cell and cell–phage interactions controlling larger scale population and community dynamics. Our system consists of dual culture biofilms of *Escherichia coli* and *Vibrio cholerae* under exposure to T7 phages, which we study using microfluidic culture, high-resolution confocal microscopy imaging, and detailed image analysis. As shown previously, sufficiently mature biofilms of *E. coli* can protect themselves from phage exposure via their curli matrix. Before this stage of biofilm structural maturity, *E. coli* is highly susceptible to phages; however, we show that these bacteria can gain lasting protection against phage exposure if they have become embedded in the bottom layers of highly packed groups of *V. cholerae* in co-culture. This protection, in turn, is dependent on the cell packing architecture controlled by *V. cholerae* biofilm matrix secretion. In this manner, *E. coli* cells that are otherwise susceptible to phage-mediated killing can survive phage exposure in the absence of de novo resistance evolution. While co-culture biofilm formation with *V. cholerae* can confer phage protection to *E. coli*, it comes at the cost of competing with *V. cholerae* and a disruption of normal curli-mediated protection for *E. coli* even in dual species biofilms grown over long time scales. This work highlights the critical importance of studying multispecies biofilm architecture and its influence on the community dynamics of bacteria and phages.

## Introduction

Many organisms find refuge from threats within groups. This observation applies across scales from bird flocks and animal herds to fish schools and insect swarms [1,2]. Bacteria are no

**Data Availability Statement:** All relevant numerical data are within the paper and its Supporting Information files.

**Funding:** This work was supported by the Simons Foundation (award number 826672 to CDN), the

National Science Foundation (award number 2017879 to CDN; award number 1817352 to CDN), and the Human Frontier Science Program (award number RGY0077/2020 to CDN). JBW is supported by a GAANN Fellowship from Department of Biological Sciences at Dartmouth. BRW is supported by a Gillman Fellowship from the Department of Biological Sciences at Dartmouth. The funders had no role in study design, data collection and analysis, decision to publish, or preparation of the manuscript.

**Competing interests:** The authors have declared that no competing interests exist.

exception and routinely live as collectives either free-floating or adhered to surfaces. Usually termed biofilms, these bacterial communities are abundant in natural settings [3–10], as are the threats faced by biofilm-dwelling microbes, such as invading competitors [11,12], diffusible antimicrobial compounds [13], phages [14,15], and predatory bacteria [16–18]. While biofilms formed by just a single species do occur, multispecies biofilms are thought to be more typical of microbial communities in the natural environment [19–22]. How predator–prey dynamics might change within multispecies biofilms is not well known, particularly at the cellular spatial scale of interactions that underlie large-scale patterns in biofilm-dominated microbial communities.

Previous work has shown that dual species biofilm cultures can increase, decrease, or have no measurable effect on phage susceptibility of a target host species living alongside a different, phage-resistant species [23–31]. Why do some multispecies biofilms confer increased phage protection to susceptible host bacteria, while others appear to do the opposite? The details underlying this variability in outcome are not well understood. A common feature among many previous studies on this topic is the use of bulk assay colony-forming unit (CFU) and plaque-forming unit (PFU) plating techniques from microtiter dish cultures; these tools, while highly effective for experimental throughput, by their nature provide an average result over entire biofilm populations residing on microtiter well walls. The conditions within these wells —for example, as a function of distance from the air–liquid interface—can vary substantially. An important way to expand on the foundation set by prior work is to examine the cellular scale variability in biofilm structure that can clarify the cell–cell and cell–phage interactions giving rise to patterns at larger spatial scales. In this paper, we target this less-explored element of phage–host interaction in multispecies contexts.

Our model system comprises dual culture biofilms of *Escherichia coli* and *Vibrio cholerae* under exposure to T7 phages or λ phages. Beyond the experimental tractability that makes *E. coli* and *V. cholerae* excellent for controlled experiments, these species can be found in natural environments together: for example, residing in brackish water [32,33] and within surface-fouling biofilms in coastal waters near human populations [34]. Members of the *Escherichia* and *Vibrio* genera are also common components of zebrafish microbiota [35,36]. Their tracta-bility make *E. coli* and *V. cholerae* together a superb platform for exploring principles of how multispecies biofilm structure could influence bacteria–phage interactions in fine detail. The cellular arrangement and secreted matrix architectures of *V. cholerae* have been explored in great detail in the last decade [37–47]. In *V. cholerae*, biofilm structure is characterized by tight cell packing coordinated by 4 matrix components: the proteins RbmA, RbmC, Bap1, and the polysaccharide VPS [46]. *E. coli* biofilms, likewise, have been dissected extensively [48–53]. T7 phages are obligately lytic and routinely isolated from the environment alongside *E. coli* [54]. T7 was used as our primary model phage, but we tested the generality of our core results with temperate phage λ as well.

Recent work has documented protection of biofilm-dwelling bacteria against phage expo-sure among several species, including *V. cholerae*, *E. coli*, *Pseudomonas aeruginosa*, and *Pan-toea stewartii* [41,55–57]. In each of these cases, phage protection has either been directly or indirectly traced to biofilm architecture controlled by secreted matrix materials. Most perti-nently, recent work in *E. coli* has shown that mature biofilms are able to block phage diffusion in a manner dependent on secreted curli polymers controlling cell–cell packing on the biofilm periphery [49,55]. Curli, along with the polysaccharide cellulose, are central elements of the *E. coli* biofilm matrix; both are commonly produced by environmental isolates of *E. coli* [52,53,58]. Phages trapped in the outer biofilm layers remain at least partially viable and can infect newly arriving susceptible bacteria colonizing the biofilm exterior [59]. In general, there is little known about how growing in a multispecies context alters biofilm matrix production

and architecture relative to that found in mono-species contexts; likewise, there is little known about whether and how these potential changes in biofilm architecture influence the ability of phages to access their hosts.

Here, we explore these open questions, studying how co-culture with *V. cholerae* influences matrix secretion and biofilm architecture of *E. coli*, and how these changes in turn influence the ability of *E. coli* to protect itself from phage attack in the midst of competition with *V. cholerae* for space and resources. We find that the patterns of phage infection among *E. coli* are qualitatively altered by the presence of a competing species, depending on cell group spatial structure.

## Results

### *E. coli* cells overgrown by and embedded within *V. cholerae* clusters are shielded from phage exposure

*V. cholerae* N16961 (serogroup O1 El Tor) and *E. coli* AR3110 were engineered to constitutively produce the fluorescent proteins mKO-κ and mKate2, respectively, such that they could be distinguished by fluorescence microscopy. We note that the strain background of *V. cholerae* that we use here, N16961, does not antagonize *E. coli* via Type VI secretion activity in culture conditions used for this study, which are detailed below [60,61]. The 2 species were inoculated at a 2:1 ratio of *V. cholerae* and *E. coli* into microfluidic devices bonded to glass coverslips, allowed to attach to the glass surface for 45 min, and then incubated under continuous flow of M9 minimal medium with 0.5% glucose for 48 h. Within this time frame, biofilms begin to form; however, monoculture *E. coli* biofilms have not yet produced sufficient curli matrix to prevent phage entry. This time frame was established by prior work and confirmed in our own experiments described below [55,59]. T7 phages were then introduced to the system continuously at $10^4$ per μL for 16 h; this strain of T7 contains a reporter construct causing infected hosts to produce sfGFP prior to lysis [55]. Changes in *E. coli* abundance and localization in the chamber were tracked and compared to those in equivalent biofilms without phage introduction.

Prior to phage introduction, we noted considerable variation in biofilm structure and composition across the glass substrata of our flow devices. Depending on the initial surface distribution of *V. cholerae* and *E. coli*, different regions of the devices contained cell groups of *E. coli* mostly on its own, locally mixed with *V. cholerae*, or occasionally embedded in the bottom layers of highly packed, *V.* cholerae-dominated clusters. Shortly after phage introduction, most *E. coli* cells growing on their own quickly began reporting infection and then lysed (Fig 1A and S1 Movie). Over the next 16 h, *E. coli* cells embedded on the bottom layers of *V. cholerae*-dominated cell groups largely survived phage exposure, with scattered singleton *E. coli* cells elsewhere in the chambers. These single cells persisted for as long as we continued to track the system (up to 144 h) but did not appear to be actively replicating. After 16 h in the dual species biofilms, waves of T7 infection could be seen proceeding partially into groups of *E. coli* embedded within *V. cholerae* biofilms, but a fraction of *E. coli* most often survived (Fig 1A).

To determine if the remaining *E. coli* survived in dual species biofilms because of de novo evolution of resistance to T7, we performed runs of this experiment after which all *E. coli* cells in the chamber were dispersed by agitation and tested for T7 resistance (see Methods). The frequency of T7 resistance in the surviving *E. coli* population was $10^{-5}$, roughly the same as the frequency of resistance prior to the introduction of T7 phages [62]. This outcome shows that there was little or no substantive population compositional shift due to selection for de novo phage resistance (S1C Fig). This is not particularly surprising, as the host and phage

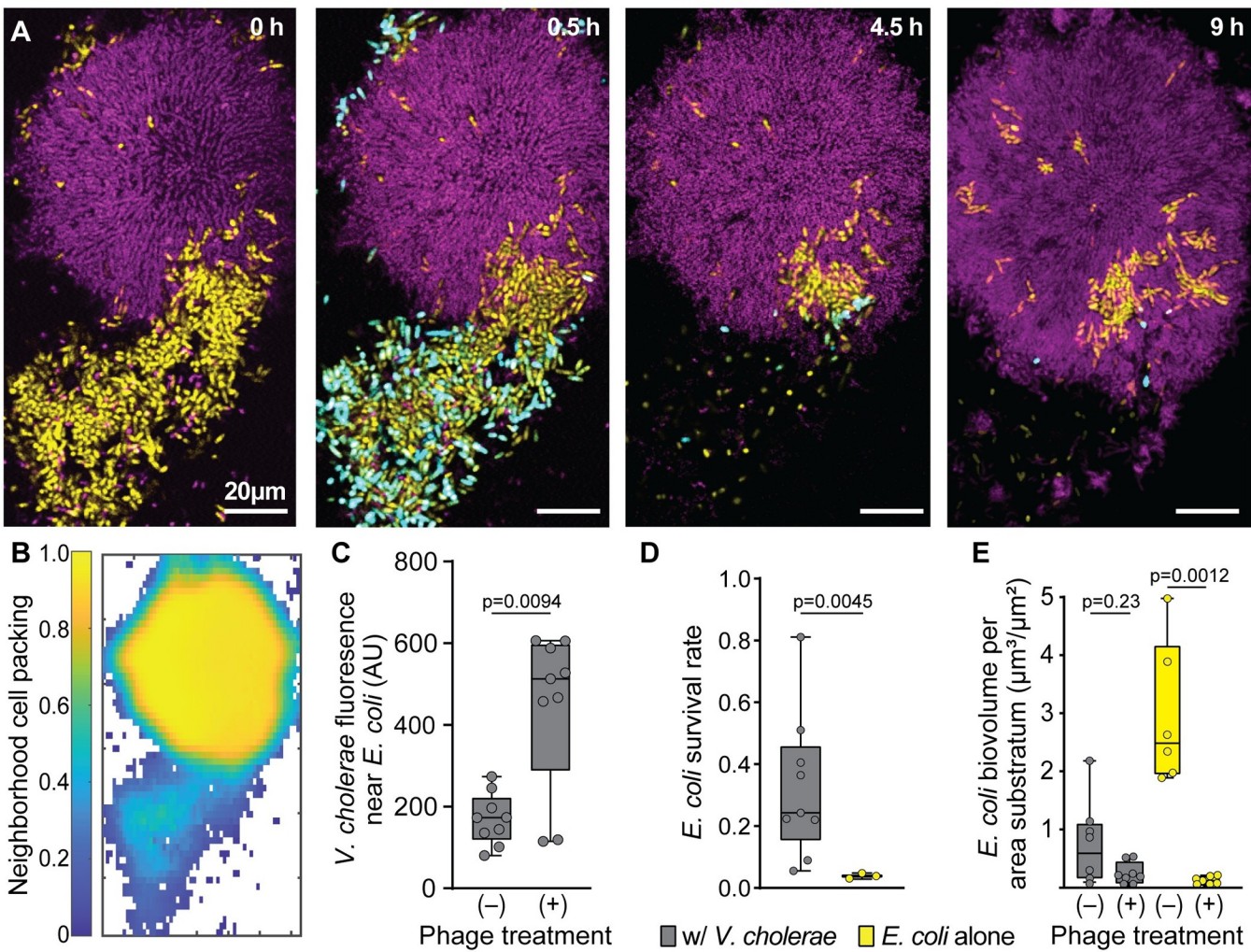

**Fig 1. *E. coli* embedded within *V. cholerae* cell groups can evade exposure to phages in the surrounding medium. (A)** Time-lapse series of a dual culture biofilm of *E. coli* (yellow) and *V. cholerae* (purple), undergoing T7 phage exposure (infected *E. coli* cells reporting in cyan/white). The biofilm was grown for 48 h prior to continuous phage introduction thereafter. Time points noted in the upper right of each panel represent time since phage introduction was started. **(B)** The neighborhood biovolume fraction (biovolume fraction within a 6 μm around each segmented bacterium) of the merged biovolumes of both *V. cholerae* and *E. coli* for the first time point in panel A. **(C)** Mean *V. cholerae* fluorescence signal found around *E. coli* cells in biofilms with and without phage exposure (Mann–Whitney U test with $n = 9$). **(D)** *E. coli* biovolume normalized to biovolume prior to the introduction of phage in dual culture with *V. cholerae* and monoculture controls (Mann–Whitney U test with $n = 9$, $n = 3$). **(E)** Total biovolume of *E. coli* in dual culture and monoculture control biofilms with and without phage exposure at equivalent time points (Mann–Whitney U tests with $n = 8$, $n = 8$, $n = 6$, $n = 7$ from left to right). The data underlying this figure can be found in S1 Data.

population sizes—and, most importantly, the extent of movement and contact events between hosts and phages—are dramatically lower in these experiments compared to those that are typical in well-mixed batch culture. Rather, these experiments suggest that T7-susceptible *E. coli* survives phage introduction in our biofilm culture conditions by avoiding exposure to them entirely when embedded in groups of *V. cholerae*. We confirmed that this outcome is specific to the biofilm context by replicating the same experiment in shaken liquid culture beginning with the same cell inoculum and phages introduced at equivalent multiplicity of infection (see Methods). In liquid co-culture conditions with *V. cholerae*, T7-susceptible *E. coli* gained no protection against phage exposure and infection (S1D Fig). In the biofilm context, the delay between the start of biofilm growth and phage introduction was important for the

experimental outcome; if phages were introduced from the beginning of biofilm growth, rather than 48 h after biofilm growth, then the extent of *E. coli* protection was all but eliminated (S2 Fig).

Our observations above suggested that in dual-species culture conditions, the majority of *E. coli* that survive phage introduction are the cell groups that have been overgrown and enveloped within the bottom layers of expanding, densely packed *V. cholerae* clusters. To test this idea quantitatively [16,63], we segmented and merged the cell volumes of *E. coli* and *V. cholerae* to calculate the joint neighborhood cell packing density for the 2 species throughout the imaged 3D space (Fig 1B). By visual inspection, regions in which *E. coli* survived contained a majority of *V. cholerae* and had relatively high cell packing (biovolume fraction >0.9), in comparison with other regions where *E. coli* tended to die of phage exposure and cell packing was lower (biovolume fraction = 0.3 to 0.6). We next measured the spatial association of *V. cholerae* with *E. coli* to see how this may change in the presence versus the absence of phage exposure. For this measurement, we segmented the *E. coli* population away from background, and then measured *V. cholerae* fluorescence in direct proximity within 2 μm of *E. coli* throughout all replicates with or without phages introduced. Compared to control experiments with no phages (Fig 1C), *V. cholerae* fluorescence was indeed significantly elevated in close proximity to *E. coli* after phage exposure, representing the surviving, protected portion of the *E. coli* population embedded within groups of *V. cholerae*. This protection effect could be replicated when introducing λ phages instead of T7 phages (S3 Fig), and in a parallel study, we show that the same effect occurs under predation by the bacterium *B. bacteriovorus* [64].

At the scale of the entire chamber community, *E. coli* showed higher survival rate in co-culture with *V. cholerae* than in monoculture on its own (Fig 1D). In absolute terms, the total population size of *E. coli* after phage exposure in co-culture with *V. cholerae* was not statistically different from the surviving population size after phage exposure in *E. coli* monoculture (Fig 1E). This result occurred because *E. coli* total abundance prior to phage introduction is lower in co-culture with *V. cholerae*, with which it is competing for space and nutrient resources, but due to embedding of many *E. coli* cells within *V. cholerae* clusters, their per-cell survival rate against phage exposure is substantially higher relative to a *E. coli* monoculture condition (Fig 1D). So, on short time scales after phage introduction (10 h), there is a significant increase in survival rate for *E. coli* growing in co-culture with *V. cholerae*, but not yet a significant difference in the absolute abundance of surviving *E. coli* relative to monoculture conditions. However, our experiments later in the paper demonstrate that on longer time scales (100+ h), the *E. coli* that survive phage exposure in co-culture within *V. cholerae* colonies maintain positive net growth and recover from the initial population decline, whereas the surviving *E. coli* cells from monoculture biofilms do not recover. Before elaborating on this point with longer time scale experiments detailed below, we first turn to the biofilm architectural mechanisms in co-culture biofilms of *E. coli* and *V. cholerae* that are responsible for the observations reported in Fig 1.

## Protection within *V. cholerae* cell clusters depends on their packing structure

After demonstrating that *E. coli* cells have reduced exposure to phages when embedded in clusters of *V. cholerae*, we explored the biofilm architectural features needed for protection to occur. As we have found previously that the extent of *V. cholerae* cell–cell packing can influence transport of phages and bacteria through biofilms, our first hypothesis based on prior work was that the high-density cell packing of *V. cholerae* biofilms was important for this protection mechanism [11,16,41]. Our other hypothesis, not mutually exclusive, was that phages

may be sequestered away from *E. coli* by irreversible attachment to the surface of *V. cholerae* cells in close proximity. To distinguish between these mechanisms, or to estimate their relative contribution to *E. coli* protection within *V. cholerae* clusters, we performed new experiments manipulating *V. cholerae* cell packing in co-culture with *E. coli* and assessing the degree of attachment and neutralization of T7 phages on the surface of *V. cholerae*.

To alter *V. cholerae* cell packing structure, we performed co-culture experiments similar to those in the previous section, but using a strain of *V. cholerae* (denoted Δ*rbmA*) with a clean deletion of the *rbmA* locus. This strain cannot produce the matrix protein RbmA, which is not essential for biofilm formation but is necessary for the tight cell packing that is characteristic of mature *V. cholerae* biofilms (S4 Fig) [44,47,65]. Biofilms without RbmA, in contrast with those of wild type (WT), can be invaded into their interior by planktonic competitor bacteria as well as predatory bacteria such as *B. bacteriovorus* [11,16]. If the high cell packing to which RbmA contributes is important to the protection of *E. coli* from phage exposure, we expect that in co-culture biofilms with *V. cholerae* Δ*rbmA*, *E. coli* will be more exposed to T7 phage predation and show different population dynamics relative to control co-cultures with WT *V. cholerae*.

We grew *E. coli* and *V. cholerae* Δ*rbmA* in biofilm co-culture, introduced T7 phages after 48 h as above, and found that the *E. coli* grown in the presence of *V. cholerae* Δ*rbmA* does not exhibit population recovery after phage introduction as it does in co-culture with *V. cholerae* WT (Fig 2A). This outcome suggests that *E. coli* does not gain protection from phage exposure amidst *V. cholerae* Δ*rbmA*, and that the cell packing architecture of *V. cholerae* WT is in fact important for this protection effect. If *E. coli* is protected within *V. cholerae* WT clusters, but not within Δ*rbmA* clusters, then in a triculture experiment of *E. coli*, *V. cholerae* Δ*rbmA*, and *V. cholerae* WT, we expect a statistical shift of *E. coli* spatial association toward WT *V. cholerae* after introducing phages as *E. coli* associated with Δ*rbmA* *V. cholerae* are more often killed. We performed this triculture experiment, measuring the average distance between *E. coli* and *V. cholerae* WT, and that between *E. coli* and *V. cholerae* Δ*rbmA*, before and after phage introduction. Without the addition of phages into the triculture condition, *E. coli* cells are just as likely to be associated with WT *V. cholerae* (median distance: 0.88 μm) as they are with Δ*rbmA* *V. cholerae* (median distance: 0.69 μm) (Fig 2B and 2C). When phages are introduced, the remaining *E. coli* were significantly closer to WT *V. cholerae* (median distance: 0.59 μm) than they were to Δ*rbmA* *V. cholerae* (median distance: 3.46 μm) (Fig 2B).

The experiments above indicate that the packing architecture of *V. cholerae* WT biofilms is important for phage exposure protection of *E. coli* within them, as *E. coli* gains little if any protection from phage exposure in proximity to loosely packed *V. cholerae* Δ*rbmA*. These data do not exclude the possibility that this difference is due in part to sequestration of phages by attachment to *V. cholerae* cells, which could occur more often in WT clusters with higher density of available *V. cholerae* cell surface relative to clusters of the Δ*rbmA* strain. To help assess whether sequestration of phages by direct attachment to *V. cholerae* cell surface was important, we incubated *V. cholerae*, *E. coli*, and Δ*trxA* *E. coli* with T7 phages in shaken liquid culture, tracking the ability to recover T7 phages every 5 min for 1 h (Fig 2E). In addition to a blank media control, the Δ*trxA* *E. coli* strain was included because this strain can adsorb phages normally but undergoes abortive infection, preventing phage amplification [66]. As expected, with *E. coli* Δ*trxA* incubation, T7 PFU recovery steadily decreased until saturation at 1 h. Incubated with T7-sensitive *E. coli* WT, T7 PFU recovery initially decreased as phage adsorption occurred, followed by a rapid increase as new phages were released. Another round of latency and amplification then occurred before the 1-h stop time. Incubated with *V. cholerae*, no change in T7 PFU recovery was observed, which was identical to the blank media control for the duration of the experiment. These data suggest that T7 phages are not sequestered by adsorption to the *V. cholerae* cell surface.

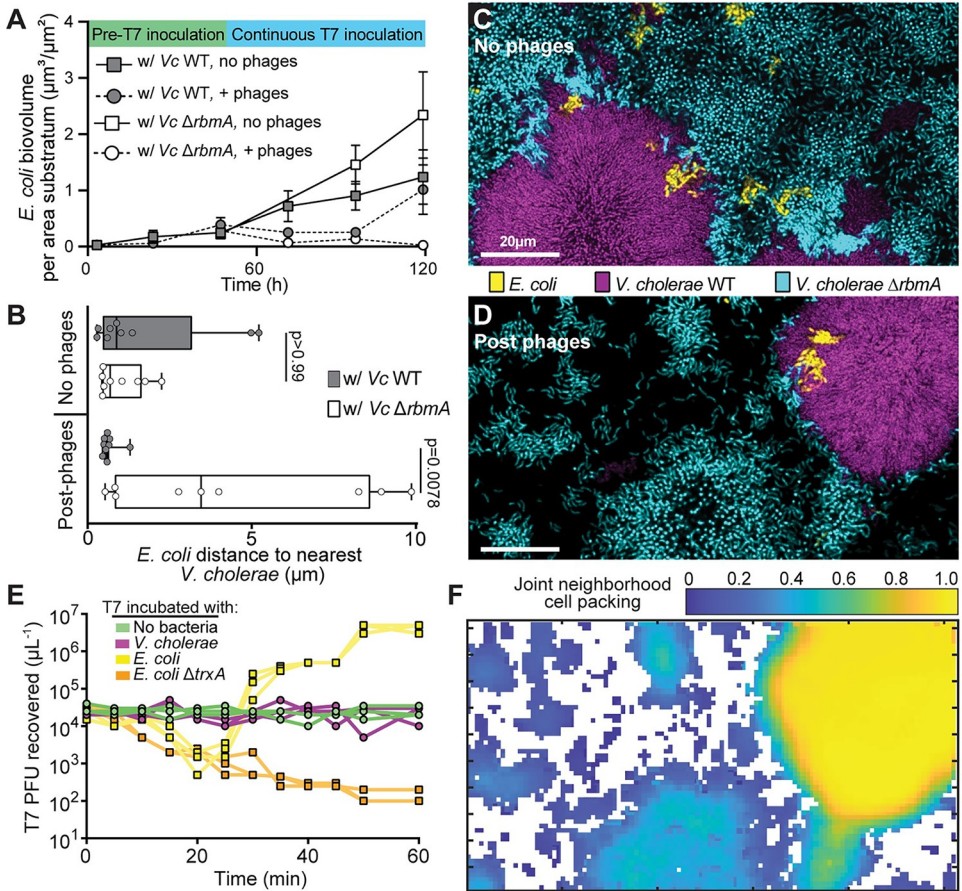

**Fig 2. *E. coli* evasion of phages within *V. cholerae* biofilms depends on the high cell–cell packing produced by WT *V. cholerae*. (A)** *E. coli* biovolume over time in dual culture conditions with either *V. cholerae* WT or *V. cholerae* *ΔrbmA* (*n* = 7, *n* = 8, *n* = 3–6, *n* = 3–8 from top to bottom in legend). **(B)** Average distance between *E. coli* cells and either *V. cholerae* WT or *ΔrbmA* in a triculture condition with or without phage exposure (Wilcoxon paired comparison tests with *n* = 9). **(C, D)** Representative images from the triculture condition with *E. coli* (yellow), *V. cholerae* WT (purple), and *V. cholerae* *ΔrbmA* (cyan) (C) without phage exposure and (D) after phage exposure. **(E)** PFU recovered after incubation of starting T7 phage inoculum with either no bacteria, *V. cholerae*, *E. coli* WT, or *E. coli* *ΔtrxA* over a 60-min time course. *E. coli* *ΔtrxA* allows for T7 phage attachment and genome ejection, but not for phage replication. Each trajectory shows the data for 1 run of each treatment (*n* = 3 for each treatment, giving 3 traces per treatment). **(F)** The neighborhood biovolume fraction of the merged biovolumes of both *V. cholerae* genotypes and *E. coli* from panel (**D**). The data underlying this figure can be found in S1 Data. PFU, plaque-forming unit; WT, wild type.

Though T7 phages do not appear to adhere to *V. cholerae* cell surface in planktonic culture, within biofilm cell clusters *V. cholerae* surface properties may differ, and they are also embedded in matrix polysaccharide and protein components. With this in mind, we also performed experiments with fluorescently labeled T7 phages in biofilm growth conditions to determine if T7 is sequestered to *V. cholerae* cell groups in this context. Labeled phages were introduced to *V. cholerae* and *E. coli* dual culture biofilms and tracked over time, and we found that they localize strongly to unprotected *E. coli* cells and not to *V. cholerae* (S5 Fig). When *V. cholerae* monoculture biofilms were grown in flow devices with labeled phages added continuously in the media, we saw no accumulation of phages along the outer surface of *V. cholerae* cell clusters (S6 Fig). We found occasional phages within *V. cholerae* cell groups along the basal glass substrata, but not in the rest of their interior volume (S6 Fig). As phages were added from the

beginning of biofilm growth onward in this experiment, the results suggest that *V. cholerae* biofilm colonies expanded over the top of initially glass-attached phages, rather than phages diffusing through biofilms to the basal layer.

Taken all together, the data in the experiments above suggest that *E. coli* is unexposed to phages within WT *V. cholerae* biofilms due to their architectural features, with minimal if any sequestration of phages by direct adsorption to the surface of *V. cholerae* cells.

## Cohabitation with *V. cholerae* alters *E. coli* matrix production

As noted in the first Results section, *E. coli* accumulates less quickly in co-culture with *V. cholerae* than it does on its own, owing to competition for limited space and resources. Previous work has shown that, in monoculture, *E. coli* biofilms can protect themselves against phages once they begin to produce curli matrix proteins, which interrupt phage binding on the single-cell scale and contribute to biofilm architecture that blocks phage diffusion on the collective cell scale [55]. Curli production does not usually start until several days after beginning *E. coli* biofilm growth in microfluidic culture conditions [55], and we wondered if growing together with *V. cholerae* in dual culture might delay or disrupt curli formation. We note again that in the experiments in previous sections, biofilms were cultivated for too short a time for *E. coli* to begin producing curli matrix even in monoculture conditions. Here, we explored whether co-culture with *V. cholerae* impacts curli production on longer time scales, when *E. coli* on its own would ordinarily be able to protect itself against phage exposure via curli production.

If curli production is reduced or disrupted by growth with *V. cholerae* as a competitor, we would expect no difference in phage exposure survival between *E. coli* WT and a strain lacking curli matrix in co-culture with *V. cholerae*. To explore this possibility, *E. coli* WT and an isogenic curli null deletion strain (denoted Δ*csgA*) were grown either on their own or in co-culture with *V. cholerae* for 96 h. This cultivation period is twice as long as is normally required for monoculture *E. coli* WT biofilms to produce curli and block phage diffusion. Biofilms were imaged at 96 h, exposed to phages at $10^4$ per μL under 0.1 μL/min flow for 16 h, and then imaged again to document population sizes of WT and Δ*csgA* before and after phage introduction. As expected, the *E. coli* WT monoculture biofilms had the highest level of survival, with some replicates showing net increases in population size after the 16-h phage treatment. *E. coli* Δ*csgA* monoculture biofilms, lacking any protection mechanism against phage exposure, had the lowest level of survival. In contrast, when in co-culture with *V. cholerae*, *E. coli* WT and Δ*csgA* (Fig 3A) showed no substantial difference in survival to phage exposure (pairwise test not significant with Bonferroni correction), suggesting that curli production is no longer necessary for T7 exposure protection for WT *E. coli* in this context.

To assess why curli-based phage protection was no longer operating for *E. coli* even in co-culture biofilms that had grown over 96 h, we repeated the experiments above with an *E. coli* WT strain harboring reporter fusions for monitoring *csgBAC* transcription and curli protein production. The transcriptional reporter was made previously by introducing *mKate2* in single copy on the chromosome within the *csgBAC* operon encoding 2 subunits of curli fiber protein (CsgB baseplate and CsgA primary curli monomer) and CsgC, which inhibits improper aggregation of CsgA monomers [55]. The protein production reporter was also made previously by introducing a 6x-His fusion tag to *csgA*, which allowed for in situ immunostaining of curli fibers produced by *E. coli* during growth in monoculture and co-culture with *V. cholerae*. As noted previously, the total population size of *E. coli* in biofilms with *V. cholerae* is lower than that found in monoculture (Fig 3B and 3E and 3F). On a per cell basis over the entire chambers, *csgBAC* transcription and curli immunostaining were significantly higher for *E. coli* growing alone versus *E. coli* growing in co-culture with *V. cholerae* (Fig 3C and 3D). These

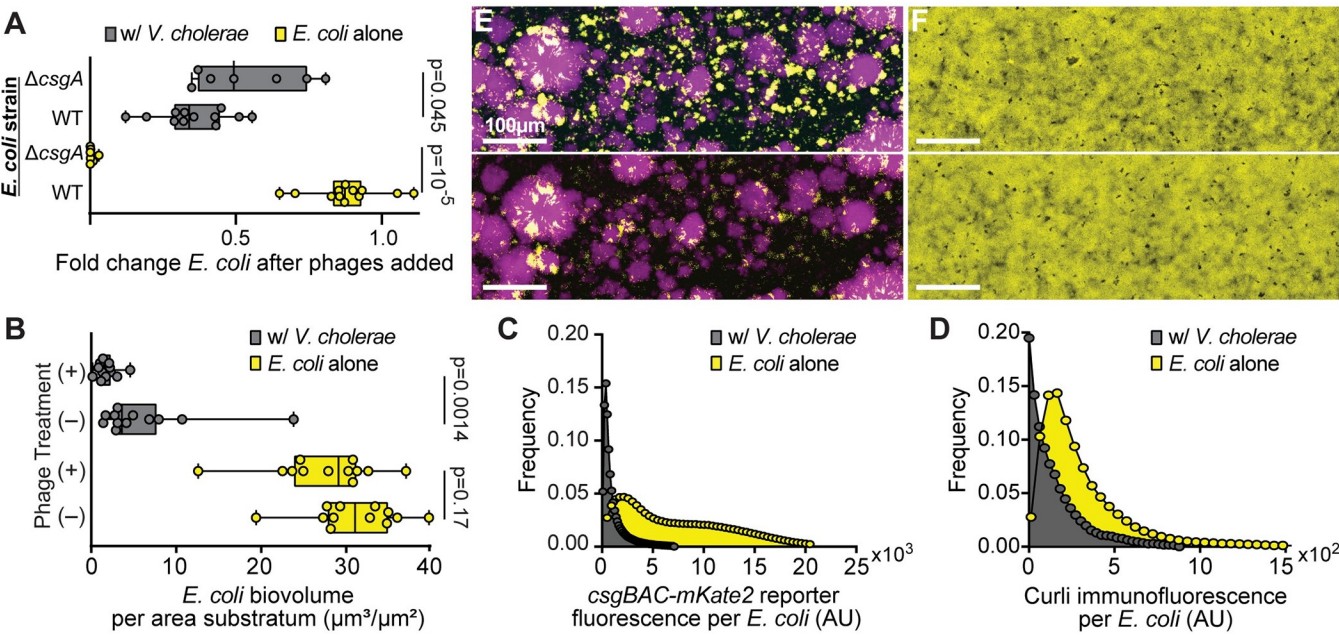

**Fig 3. *E. coli* biofilms' normal production of curli matrix protein is interrupted in co-culture with *V. cholerae* to the extent that phage protection is no longer provided by *E. coli* biofilm matrix.** (A) *E. coli* biovolume normalized to biovolume prior to phage introduction in dual culture and monoculture conditions for both *E. coli* WT and *E. coli* Δ*csgA* (Mann–Whitney U tests with $n = 7$, $n = 12$). (B) Total *E. coli* biovolume with and without phage treatments at equivalent time points (Mann–Whitney U tests with $n = 12$). In these experiments, in contrast with Fig 1E, biofilms were grown for longer periods before phage addition such that *E. coli* WT on its own could produce protective curli matrix prior to phage addition. (C) Frequency distribution of *csgBAC* transcriptional reporter fluorescence around *E. coli* in monoculture and dual culture conditions. (D) Frequency distribution of curli immunofluorescence intensity in proximity to *E. coli* in monoculture and dual culture conditions. (E) Dual culture conditions of *E. coli* (yellow) and *V. cholerae* (purple) before phage exposure (top) and after 16 h of continuous phage exposure (bottom). (F) Monoculture conditions of *E. coli* before phage exposure (top) and after phage exposure (bottom). The data underlying this figure can be found in S1 Data.

patterns manifested at the scale of the whole chamber; on a smaller spatial scale, *E. coli* distance from *V. cholerae* in co-culture was not correlated with curli production (S7 Fig).

Overall, these results suggest that *E. coli* curli production is substantially reduced when growing together with *V. cholerae*. It is not clear exactly why this is the case, but we speculate here that co-culture with *V. cholerae* alters one or a combination of nutrient availability, microenvironment osmolarity, and envelope stress experienced by *E. coli*, all of which influence the regulation of curli production [52]. The reduction in curli production may in turn contribute to the loss of curli-based protection against T7 phages even after long incubation periods over which *E. coli* normally develops curli-based phage protection on its own in monoculture. Together with the previous section, our experiments here also indicate that while *E. coli* has a lower ability to protect itself via curli matrix production when in co-culture, it can avoid phage exposure altogether when it been overgrown by and embedded within *V. cholerae* colonies.

## Ecological consequences of joint interspecific competition and phage exposure

Our results thus far suggest complex ecological dynamics in which *E. coli* suffers a fitness reduction in spatial competition with *V. cholerae*, but on the other hand, *E. coli* gains a protective fitness benefit against phage exposure when embedded in the highly packed biofilm cell clusters that *V. cholerae* produces. It is still not clear, though, whether this protection is lasting under prolonged phage exposure, or whether *E. coli* remains viable within *V. cholerae* clusters despite being packed into their bottom-most cell layers. To characterize these population

dynamics more thoroughly, we performed new experiments in which *E. coli* and *V. cholerae* were inoculated alone or together and grown for 48 h, followed by either continuous phage exposure or no phage exposure for an additional 96 h (Fig 4). Note that this experimental regime is such that even in monoculture, *E. coli* will not have produced sufficient curli to block phage diffusion at the onset of phage influx into the biofilm chambers [55].

The population dynamics of *E. coli* and *V. cholerae* without T7 phage addition confirm our earlier suggestion that this interaction is competitive by default; *E. coli* population size is reduced in co-culture relative to when growing on its own (Fig 4E; yellow versus gray square trajectories). *V. cholerae* total productivity is also reduced in co-culture with *E. coli* relative to when growing on its own (Fig 4F), though overall it outcompetes *E. coli* by a substantial margin (S8 Fig). This result was driven by *V. cholerae* biofilm clusters expanding more rapidly and robustly in lateral and vertical space, displacing some neighboring *E. coli* and overgrowing other *E. coli* cell groups along the glass surface (Fig 4B). Under prolonged phage exposure, however, these same enveloped clusters remain mostly protected from phage killing. We did observe occasional *E. coli* deaths within trapped clusters, shown via the T7 infection reporter (Fig 4C and 4D), but overall, the *E. coli* cell groups maintained positive net growth and expanded laterally as the overlaid *V. cholerae* biofilms expanded as well (Fig 4E). Based on our earlier experiments, we suspect that progeny phages released from these occasional infection events within protected clusters were mostly trapped in place, and a sufficient impedance to phage diffusion allows for long-term survival of phage-susceptible hosts in close proximity [55,67–69]. Though we have linked *V. cholerae* cell packing to this phage diffusion limitation, the exact biophysical explanation for limited phage diffusion is an important future question. We speculate here that high density packing of *V. cholerae*, combined with the biochemical properties of its matrix and sequestration of phages to trapped debris from lysed *E. coli*, all contribute to the strongly impeded diffusion of T7 phages from initial sites of infection and amplification.

Given that monoculture *E. coli* biofilms have a lower cell packing density than *V. cholerae* biofilms (S9 Fig), and that the inclusions of *E. coli* within the *V. cholerae* biofilms continue expanding through time, we were curious to see if *E. coli* cell groups trapped within *V. cholerae* biofilms interrupted their highly packed structure. We assessed this question by calculating their 2 species' joint neighborhood cell packing, finding it to be stable over time and indistinguishable from what *V. cholerae* produces on its own. This suggests that the *V. cholerae* biofilm architecture, once it has been initiated by *V. cholerae* cells growing together, can drive cell groups of other species trapped within them into high packing orientations that do not disrupt the overall structure (S9 Fig) [40,43,70,71]. As we explored in a parallel study on *B. bacteriovorus* predation in dual species biofilms, co-culture with *E. coli* only disrupts *V. cholerae* architecture when cells of both species begin dividing directly adjacent to each other from the outset of biofilm growth [64].

From an ecological point of view, the net result of these architectural details is that if phage exposure occurs before *E. coli* is able to produce protective curli matrix, *E. coli* has higher absolute fitness in co-culture with *V. cholerae*—with which it is otherwise competing—than it does on its own (Fig 4E, yellow versus gray circles). The same process by which *E. coli* is overgrown and enveloped by expanding *V. cholerae* biofilms, which in the absence of phages reduces *E. coli* population growth relative to monoculture, protects *E. coli* from near total population collapse when phages are present. Since *V. cholerae* still has somewhat reduced absolute fitness in co-culture with *E. coli* compared to monoculture (regardless of phage addition: Figs 4F and S8), this interaction can be characterized as *E. coli* parasitizing or exploiting *V. cholerae* biofilm structure and gaining some protection at their expense in the presence of *E. coli*-targeting phages.

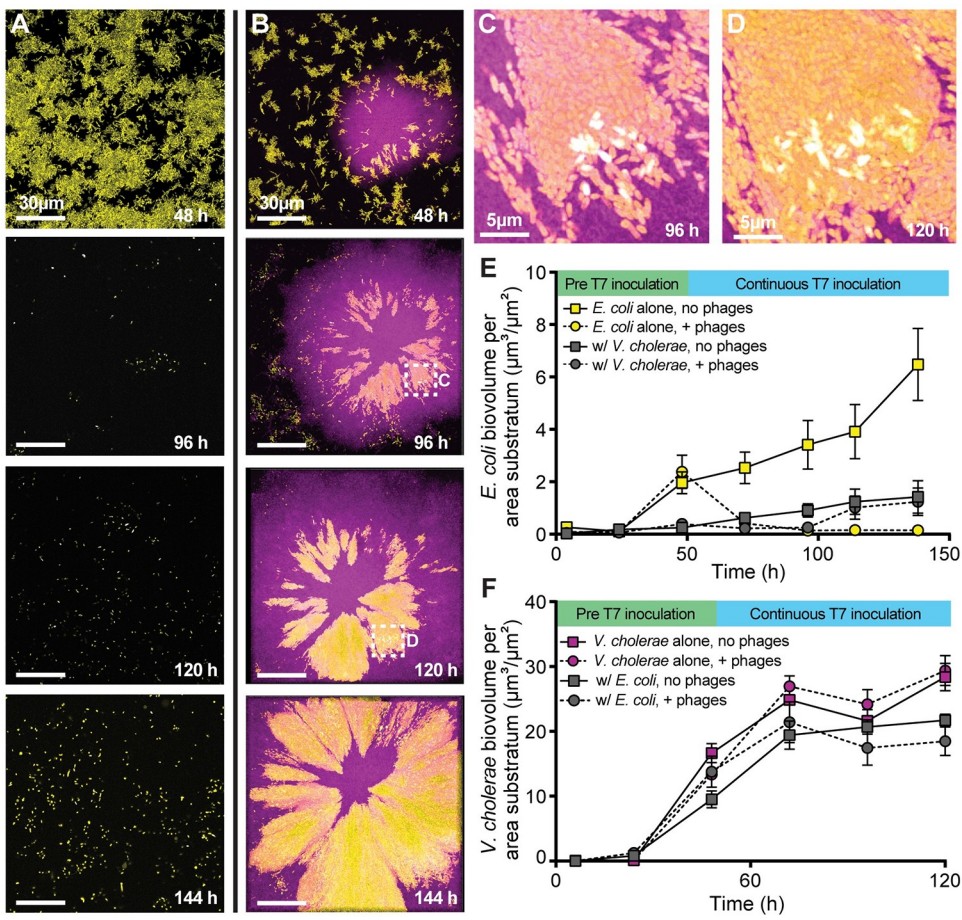

**Fig 4. Population dynamics of *E. coli* (yellow) and *V. cholerae* (purple) in monoculture and dual culture conditions, where biofilms grew for 48 h prior to phage exposure, and phage exposure was applied continuously for 96 h thereafter.** (**A, B**) Representative images from time course imaging of (A) *E. coli* monoculture and (B) co-culture with *V. cholerae*. (**C, D**) Magnification of *E. coli* phage infection (reporting in cyan/white) within a cluster embedded in a larger colony of *V. cholerae* at (C) 96 h and (D) 120 h. Expanded fields of view in (C) and (D) are denoted by checked white boxes in panel B. (**E**) *E. coli* population dynamics in monoculture and in co-culture with *V. cholerae*, with and without T7 phage exposure from 48 h onward ($n = 7$, $n = 8$, $n = 6–7$, $n = 6–8$ from top to bottom in the legend). Note that the data in the gray circles and squares through 120 h are repeated from the gray data in Fig 2A. (**F**) *V. cholerae* population dynamics in monoculture and in co-culture with *E. coli*, with and without T7 phage exposure from 48 h onward ($n = 4$, $n = 4$, $n = 4–8$, $n = 4–8$ from top to bottom in the legend). The data underlying this figure can be found in S1 Data.

## Discussion

How cell group architecture in biofilms influences bacterial community dynamics, and vice versa, are important questions in microbial ecology that will benefit from recent advances in live microscopy and image analysis [22,63,72–74]. Here, we have explored the spatial population dynamics of *E. coli* cohabiting biofilms with *V. cholerae*, asking in particular how this dual species system influences the interaction between *E. coli* and the lytic phage T7. *E. coli* biofilms can self-protect against T7 phage exposure by producing curli matrix, but before producing curli, *E. coli* biofilms are highly vulnerable to phages [55]. When otherwise *E. coli* populations would collapse due to T7-mediated killing, they benefit from biofilm co-habitation with *V. cholerae*. This occurs because pockets of *E. coli* are overgrown and enveloped within densely packed, laterally expanding *V. cholerae* cell clusters whose structure greatly reduces phage

diffusion. We identified the packing structure of *V. cholerae* biofilms as essential to T7 phage blocking, as has been implied previously for *Vibrio* phages as well [41]. There are likely other contributions toward phage blocking from the electrostatic and hydrophobicity properties of *V. cholerae* biofilm matrix [75], which are notable questions for future work. We demonstrated that *E. coli* matrix production is altered in longer term biofilm co-culture with *V. cholerae*, with which, by default, *E. coli* competes for space and resources. Interestingly, with phages introduced to the system, this relationship becomes parasitic/exploitative on the part of *E. coli*, which gains protection from phage exposure while taking up space that *V. cholerae* could ordinarily occupy within the highly packed cell groups that it produces. These observations emphasize the importance of carefully observing the distinctions between single and multispecies biofilm architectural development, which in turn impact how phage–bacteria infection dynamics occur in multispecies contexts. The molecular mechanisms underlying differences between single species and multispecies biofilm architectures remain underexplored, as do the implications for biofilm ecology and microbial community ecology more generally.

Prior literature has documented that co-habiting biofilms with other microbial species can render bacteria more, less, or equally susceptible to phage attack in comparison to when growing in single species conditions under phage exposure [31]. Our experiments here document that all of these outcomes can occur within the same system occupying less than 1 mm$^2$, depending on the detailed biofilm architecture of the 2 species and the timing of biofilm growth before exposure to phages. For example, in biofilms grown for less time that *E. coli* needs to produce curli matrix and protect itself from phages, co-culture with *V. cholerae* leads to increased phage protection in locations where *E. coli* becomes overgrown and embedded within *V. cholerae* clusters. But there is no change in vulnerability in locations where *E. coli* grows in clusters just outside the periphery of tightly packed *V. cholerae* groups, which may be directly adjacent to locations inside the *V. cholerae* groups where *E. coli* is protected (Fig 1). On the other hand, in biofilms grown for longer periods over which *E. coli* would normally protect itself via curli production on its own, co-culture with *V. cholerae* leads to reduced curli production and reduced protection from phage exposure for any *E. coli* not embedded in *V. cholerae* cell clusters (Fig 3).

Our work provides a biofilm-specific context directly connected to the idea of phage protection via spatial refuges that has been explored in the phage ecology literature [68,69,76–80]. Such refuges, even when transient, can be sufficient to support coexistence between phages and susceptible host bacteria [62,81]. Our work adds mechanistic insight into how spatial refuge-based phage protection can depend on the nuances of biofilm architecture, which in turn are distinct in multispecies versus mono-species contexts. Another important implication of our results is that the relative frequencies of different species and their initial surface colonization densities can cascade into differences in distribution of, for example, mixed-species versus mono-species biofilm architecture, which in turn can strongly influence the survival of susceptible bacteria to phage exposure. Phage exposure can then systematically shift the community architectural composition—for example, in our case, by eliminating any cell groups of *E. coli* that are not embedded within highly packed biofilm clusters of *V. cholerae* [79].

The data presented here provide a proof of principle that multispecies biofilm structure can provide protection to otherwise phage-susceptible bacteria, and that this protection depends on the cellular resolution details of biofilm architecture. There are important caveats, however. Though *V. cholerae* and *E. coli* can be found in the same environmental locations in proximity to human populations, they are not necessarily frequent biofilm co-habitants in nature. Our microfluidic flow conditions, though they capture some essential environmental features of biofilm growth for many species, are simplified relative to the diverse natural settings in which surface adherence and matrix production occur. In future work, it will be vital to explore

systems with increased ecological realism in terms of species composition and environmental topography while sacrificing minimal tractability for live imaging. This pursuit will also be important for determining how *E. coli* biofilm architecture, including curli production and its contribution to phage protection, depends on environmental conditions and co-habiting community members in habitats with as much realistic detail as possible.

With these limitations in mind, we take note of 2 core observations here, namely that (1) *E. coli* embedded in *V. cholerae* biofilm cell groups can avoid and survive phage exposure where otherwise they would be exposed and killed, and that (2) this phage protection can be eliminated by the deletion of a single matrix gene that loosens the packing architecture of *V. cholerae*. Taken together, these observations strongly suggest that the extent to which multispecies biofilm architecture influences phage–bacteria population and evolutionary dynamics in nature will depend on the particular species involved in the microbial community in question, their respective biofilm architectures in mono-species and multispecies contexts, and the dependence of these architectures on local environmental conditions. Our study thus emphasizes that it is crucial to examine more examples of biofilm communities and the dynamics of phage–bacteria encounters within them using high-resolution live imaging techniques. We expect that studying other systems of multi-host, multi-phage composition at this level of spatial detail will reveal similarly complex connections between community ecology, the nuances of cell group architecture, and the time scales of biofilm growth versus phage exposure. This future work will be important not only for understanding fundamental microbial natural history, but also for defining the contingencies under which phage applications for antimicrobial therapy might be hindered by the presence of nontarget species cohabiting with biofilm-producing pathogens.

How spatial constraints influence community ecology has gained momentum as an important frontier in microbiology research as we try to relate the massive amount of sequencing data on community composition to the cellular scale processes of multispecies interaction [21,22,82–84]. This study examines what is possible when 2 species that construct biofilms with different combinations of cell growth pattern and matrix composition interact together under phage exposure for one of the bacterial biofilm inhabitants. Future work will benefit from ever increasing realism in the species composition and environmental features with which the many elements of phage ecology within biofilms can be explored.

## Methods

### Strains

All *V. cholerae* strains used in this study are strain N16961 (serogroup O1 El Tor) and derivatives (Table 1). The fluorescent protein expression construct insertions and Δ*rbmA* deletion mutants were made here and previously using standard allelic exchange [41,85]. *E. coli* strains are all AR3110 and its derivatives. AR3110 was derived from the K-12 strain W3110, for which cellulose production is disrupted by a polar stop codon mutation in *bcsQ*. This stop codon was corrected in AR3110 to yield a strain that produces the full complement of *E. coli* biofilm matrix components including cellulose and curli protein [48]. Like other K-12 derivatives, the *E. coli* AR3110 parental strain lacks O-antigen and is susceptible to T7 phages. AR3110 derivatives were produced via lambda red recombination or through allelic exchange. Briefly, primers encoding regions of homology to the host genome were used to amplify fluorescent protein expression constructs fused by SOE PCR to Kan$^R$ or Cm$^R$ resistance cassettes. These PCR products were used to knock in the fluorescent marker and selected for with the respective resistance cassette [55,59]. Recombinant T7 phages were created previously using T7select415-1 phage display system [55]. Recombinant λ phages were generously provided by Lanying

**Table 1. Bacterial strains and reagents used in this study.**

| Strain | Relevant markers/genotype | Source |
|---|---|---|
| *V. cholerae* | | |
| CNV 121 | N16961, rbmA-3xFLAG, LacZ::Ptac-mKO-κ | [85] |
| CNV 126 | N16961, ΔrbmA, LacZ::Ptac-mKo-κ | [41] |
| CNV 248 | N16961, LacZ::Ptac mTFP | This study |
| *E. coli* | | |
| CNE 320 | AR3110, with ΔcsgA::scar, Ptac-mRuby2 and KanR inserted at attB site | [55] |
| CNE 336 | AR3110, with csgBAC-mKate2 transcriptional fusion, Ptac-mKO-κ and KanR inserted at attB site | [55] |
| CNE 761 | AR3110, *attB::mKate2-KanR* | [59] |
| CNE 772 | AR3110, *attB::mKate2-KanR, ΔtrxA* | [59] |
| CNE 776 | AR3110, *attB::mKate2-KanR, 6x-His csgA* | [59] |
| **Phages** | | |
| CNX 9 | WT phage T7 | DSMZ (DSM4623) |
| CNX 11 | T7 with sfGFP under control of phi 10 promoter | [55] |
| CNX 19 | *λD-mNeongreen cI$_{857}$-mKate2 bor::Cm$^R$* | [86] |
| **Chemicals and reagents** | **Source** | **Product number** |
| Kanamycin | Millipore-Sigma | cat.#60615 |
| MEM Vitamin Solutions | Millipore-Sigma | cat. #M6895 |
| Alexa Fluor 633 NHS Ester | Thermo Fisher Scientific | cat. # A20005 |
| Anti-6X His Epitope Tag (Rabbit) antibody conjugated to Dylight 405 | Rockland Immunochemicals | cat. # 600-446-382 |
| Poly-dimethysiloxane (PDMS) | Dow Chemical Company; SYLGARD 184 | cat. # 04019862 |
| #1.5 glass coverslips | Azer Scientific | cat. # 1152260 |
| Inlet tubing | Cole Parmer | cat. # 06417–11 |
| 27Gx1/2 needles | BD Precision | cat. # 30510 |
| 1 mL syringes | Brandzig | cat. #CMD2583 |
| Harvard Apparatus Pico Plus Elite syringe pumps | Harvard Apparatus | cat. # 70–4506 |
| **Software and algorithims** | **Source** | **Version** |
| Zen Black | Zeiss | v14.0.0.0 |
| Zen Blue | Zeiss | v3.4.91.00000 |
| MATLAB | MathWorks | vR2021a |
| Paraview | Kitware | v5.9.1 |
| Prism | GraphPad | v9.4.1 |
| BiofilmQ | [61] | v0.2.2 |

Zeng's group, which created them by infecting *λDam cI$_{857}$ bor*::*KanR* phages on LE392 (permissive host) with plasmid pBR322-λD-mTurquoise2/mNeongreen-E for recombination, and further selection for fluorescent plaques [86].

## Microfluidic flow device fabrication

Microfluidic chambers are produced by casting poly-dimethysiloxane (PDMS; Dow Chemical Company, SYLGARD 184, cat. # 04019862) onto preexisting chamber molds (a diagram of the chamber design used in this study is provided in S10 Fig). The resulting PDMS blocks were cut to size, hole-punched for inlet and outlet channels, and then bonded to #1.5 glass coverslips using plasma cleaning preparation of the PDMS and glass coverslips (Azer Scientific, cat. # 1152260). Between the inlet and outlet port areas, the internal space of the chambers in which

biofilms were cultivated measured 5,000 μm × 500 μm × 70 μm (LxWxH). Segments of inlet tubing (Cole Parmer PTFE #30, cat. # 06417–11) attached to 27Gx1/2 needles (BD Precision, cat. # 305109) on 1 mL syringes (Brandzig, cat. #CMD2583) were plumbed into chamber inlets, and the syringes were driven by Harvard Apparatus Pico Plus Elite syringe pumps (Harvard Apparatus, cat. # 70–4506). Tubing from chamber outlet channels was fed to effluent collection dishes.

## Biofilm culture conditions

Overnight cultures of *V. cholerae* and *E. coli* were inoculated into the microfluidic chambers at a ratio of 2:1. To achieve comparable amounts of biofilm growth in the *ΔrbmA V. cholerae*, WT *V. cholerae*, and *E. coli* triculture experiments, chambers were inoculated with a ratio of 6:3:1, respectively. After a 45-min incubation period without flow to allow for surface attachment, M9 minimal media with 0.5% glucose continuously flowed into the chamber at a rate of 0.1 μL/min. For experiments with longer term incubation to promote curli production by *E. coli* prior to phage introduction, and to discourage biofilm overgrowth of the chamber, chambers were incubated with M9 minimal media with 0.25% glucose.

For the immunostaining of curli, a new AR3110 strain harboring a translational 6xHis tag fused to *csgA*, which encodes the monomer for curli production, was stained with Anti-6X His Epitope Tag (Rabbit) antibody conjugated to Dylight 405 (Rockland Immunochemicals, cat. # 600-446-382) added to the media at a concentration of 0.1 μg/mL for the entirety of the experiment. Prior work has shown that addition of the 6xHis tag to CsgA does not interrupt its function by any measures tested [55]. For experiments investigating the effects of phage exposure, pairwise comparisons were made between flow devices of equal age that had phages added or did not have phages added, as the control for these experiments was the dynamics of co-culture growth without the presence of phage. For experiments investigating the disruption of curli matrix production, comparisons were made between flow devices before they experience phage exposure and after they experience phage exposure, as the control for these experiments were *E. coli* monoculture biofilms surviving exposure from phages. All experiments were carried out at room temperature.

## Phage propagation, staining, and introduction to biofilms

T7 phages were produced by growing sensitive *E. coli* to $OD_{600}$ = 0.4 in M9 minimal media with 0.5% glucose, before adding an aliquot of T7 phage and incubating until the bacterial cultures were cleared. Phages were quantified using standard plaquing techniques and back diluted to $10^4$/μL in M9 with 0.5% glucose. To visualize phage infection, we used a previously constructed T7 strain that induces sfGFP production by the host prior to lysis; to visualize phages directly, we stained T7 phages with Alexa Fluor 633 NHS Ester (Thermo Fisher Scientific, cat. # A20005) using the Phage on Tap protocol [87]. For experiments in which capsid-labeled phages were introduced to biofilms, the biofilms were grown for 48 h as described above, followed by continuous labeled phage addition at $5 \times 10^6$/μL for the remainder of the experiment. λ phages were produced by growing lysogenic *E. coli* to an $OD_{600}$ = 0.2 in M9 minimal media with 0.5% glucose, then heat shocked at 42˚C for 20 min, and then incubated at 37˚C until visible lysis occurred. For experiments with short phage exposure, phages were continuously introduced for 16 h. For experiments with extended phage exposure (Fig 4), phages were continuously added for 96 h. For the nascent biofilm phage exposures, phages were introduced into the chamber immediately following the initial attachment step. For the labeled phage time lapse, phages were added for a total of 20 h.

## Biofilm dispersal and detection of de novo T7-resistance mutants from biofilm culture

*V. cholerae* and *E. coli* dual culture biofilms grown for 48 h and then treated with phages for 16 h were dispersed by removing the tubing from the microfluidic device and vigorously pipetting 100 μL of M9 media and air bubbles back and forth between the inlet and outlet ports. This was done to ensure maximal removal of all cells in the chamber in order to capture accurate measurements of total *E. coli* and de novo T7-resistant *E. coli*. To determine cell viability and phage sensitivity, this 100 μL volume containing dispersed biofilm cells was serially diluted and plated on LB+50 μg/mL kanamycin plates for total *E. coli* counts and, in parallel, on LB +50 μg/mL kanamycin plates saturated with T7 phages to determine de novo T7-resistant counts. Kanamycin was added to selectively plate for *E. coli* and kill *V. cholerae*, as all *E. coli* strains carried a kanamycin resistance cassette on the chromosome from insertion of their constitutive fluorescent protein production constructs [55].

## Dual liquid culture phage assay

Approximately 5 mL liquid dual cultures of room temperature M9 minimal media with 0.5% glucose were inoculated 2:1 with *V. cholerae* and *E. coli* each normalized to $OD_{600}$ = 0.15. A total of 5 μL samples of these dual cultures were taken at regular time intervals, serially diluted, and plated onto LB+50 μg/mL kanamycin plates to obtain the CFU count for *E. coli*. At an $OD_{600}$ measurement of 1.5 (15 h), $6.5 \times 10^7$ T7 phages were introduced to the dual cultures, as this is the estimated MOI for phages introduced in the biofilm condition. Note that MOI (multiplicity of infection) estimation is less straightforward in biofilm culture, as many phages introduced by flow do not contact host cells and pass out of the chambers in the liquid effluent.

## Phage adsorption assay

Bacterial cultures of *E. coli*, *E. coli* Δ*trxA*, and *V. cholerae* were grown and back-diluted to an $OD_{600}$ = 0.5 in LB medium. T7 phages were added to a final concentration of $5 \times 10^4$ phages per μL. Cultures were incubated at 37°C on an orbital shaking platform, and 500 μL aliquots were taken every 5 min, passed through a 0.2-μm filter, and stored on ice until the end of the experiment. The filtration step served to exclude any bacterial cells, and any phages that were attached to them, allowing us to measure free phages remaining in the liquid medium. Flow-through samples were then serially diluted and plated for PFUs.

## Microscopy and image analysis

All imaging was performed using a Zeiss 880 line-scanning confocal microscope, using a 40x/ 1.2 N.A. water objective or a 10x/.4 N.A. water objective. The 6xHis Tag Antibody Dylight 405 that was used to stain 6xHis-tagged curli polymers was excited with a 405 laser line. The mTFP protein that *V. cholerae* produces in experiments investigating curli transcription was excited with the 458 laser line. The sfGFP protein produced by the T7 infection reporter construct and the mNeonGreen capsid label of the λ phages were both excited (in separate experiments) with a 488 laser line. The mKO-κ protein that *V. cholerae* expresses constitutively and by *E. coli* that reports *csgBAC* transcription was excited with a 543 laser line (in separate experiments). The mKate2 protein that *E. coli* expresses constitutively and upon *csgBAC* transcription, and the mRuby2 protein that ΔcsgA *E. coli* constitutively expresses was excited with a 594 laser line (in separate experiments). The Alexa Fluor 633 conjugated onto the capsid of labeled T7 phage virions was excited with a 633 laser line. For each chamber in each experiment, multiple independent locations were chosen within each biofilm chamber and averaged to give 1

measurement for a given chamber in the case of whole-biofilm measurements. Prior to export, images were processed by constrained iterative deconvolution in ZEN blue.

### Replication, quantification, and statistics

Replication is reported for each experiment individually in the legends of all of the figures. The reported *n* for each figure panel refers to biological replicates. One biological replicate was defined as the averaged outcome for measurements across a single microfluidic flow chamber inoculated from independent overnight culture preparations. Biological replicates for the core biofilm microscopy experiments in the study were performed across 2 to 3 weeks with independent microfluidic chambers. Technical replicates were separate z-stacks captured at randomized locations throughout a given flow chamber; measurements from these technical replicates were averaged to calculate the value for the biological replicate corresponding to that flow chamber. All biofilm image quantification was performed within the BiofilmQ framework [63]. For 3D grid-based measurements detailing microscale architecture, segmented microbial volumes were divided into a 3D grid with each node 0.8 μm on a side. Joint neighborhood cell packing measurements merged the biovolume of all bacteria within a sample and calculated the local biovolume fraction within 6 μm of each segmented bacterial volume within each grid cube [16]. For experiments using λ phages, infection was measured by calculating a Mander's overlap coefficient between *E. coli* cells and λ phages. Mann–Whitney U tests with the Bonferroni correction were used for pairwise comparisons. We chose nonparametric comparison tests because they are relatively conservative and because the assumptions required for parametric tests could not consistently be assessed for our data.

## Supporting information

**S1 Movie. A time lapse of a dual culture biofilm of *E. coli* (yellow) and *V. cholerae* (purple), undergoing T7 phage exposure (infected *E. coli* cells reporting in cyan/white).** The dual species biofilm was grown for 48 h prior to continuous phage introduction for the next 16 h. The video begins immediately after the start of phage introduction, and the elapsed time since phage introduction is indicated at the bottom of each frame. This is the full image sequence from which the representative images were taken for Fig 1A of the main text.
(MP4)

**S1 Data. This compressed directory contains Excel files with raw numerical data contributing to the main text and SI Figures as noted in the respective figure legends.**
(ZIP)

**S1 Fig. *E. coli* cells can survive T7 phage exposure within multispecies biofilms in the absence of de novo phage resistance evolution.** (**A**) A co-culture biofilm of *V. cholerae* (purple) and *E. coli* (yellow) after 16 h of continuous phage exposure. (**B**) The same microcolony as (**A**) after heavy disturbance to clear *E. coli* cells out of the chambers to test for phage resistance. (**C**) *E. coli* CFU recovered from co-coculture flow devices when plated without T7 phages (for total counts) or plates saturated with T7 phages (for de novo T7-resistant mutants) (*n* = 4). (**D**) *E. coli* CFU in liquid culture with *V. cholerae* over time with and without the addition of phages. The addition of *V. cholerae* in shaken liquid culture did not confer protection against phage exposure (*n* = 3). The data underlying this figure can be found in S1 Data.
(PDF)

**S2 Fig. *E. coli* (yellow) does not gain phage exposure protection in co-culture with *V. cholerae* (purple) if phages are added to the culture from the beginning of biofilm growth.** (**A, B**) We surveyed biofilms extensively to see if *E. coli* ever survived when phages were added from the beginning of biofilm growth. Sporadic *E. coli* that had survived phage exposure could

be found, but only very rarely. The image in panel (A) is one of only 3 instances out of hundreds of images in which any *E. coli* were found. (**C**) *E. coli* total abundance over time when phages are added continuously from the beginning of biofilm growth either in monoculture or with *V. cholerae* (*n* = 6–12). The data underlying this figure can be found in S1 Data.
(PDF)

**S3 Fig. *E. coli* cells can evade exposure to λ phages when embedded in *V. cholerae* cell groups in the same manner as observed for T7 phage exposure.** (**A**) 3D rendering of *V. cholerae* (purple), *E. coli* (yellow), and *E. coli* with λ phages attached to their cell surface (red). (**B**) Quantification of *E. coli* and λ phage overlap in a top-down view of the biofilm rendered in panel A. *E. coli* clusters within *V. cholerae* biofilms generally evade λ phages, as seen with T7 phages. (**C**) Frequency of phage infection, measured by Mander's overlap coefficient between *E. coli* and λ phage fluorescent signal, as a function of the *V. cholerae* fluorescence shell in proximity to *E. coli*. The data underlying this figure can be found in S1 Data.
(PDF)

**S4 Fig. Quantification of cell packing for WT *V. cholerae* (purple) and Δ*rbmA V. cholerae* (cyan) in co-culture with *E. coli* (yellow).** (**A**) Representative image of a triculture condition of Δ*rbmA V. cholerae*, *V. cholerae*, and *E. coli*. (**B**) The neighborhood biovolume fraction of the merged biovolumes of both *V. cholerae* genotypes and *E. coli* from (**A**). The data underlying this figure can be found in S1 Data.
(PDF)

**S5 Fig. Co-culture biofilms exposed to dye-conjugated T7 phages (cyan) show minimal association of phages to *V. cholerae* cell groups (purple) and high T7 localization to *E. coli* (yellow).**
(PDF)

**S6 Fig. T7 phages do not generally enter the interior or accumulate on the outer periphery of *V. cholerae* biofilms.** *V. cholerae* biofilms (purple) grown while dye-conjugated T7 phages (cyan) were continously added into the flow devices from the beginning of biofilm growth for 96 h. (**A–D**) Representative image slices taken from a biofilm (A) 1.54 μm, (B) 2.70 μm, (C) 4.25 μm, and (D) 13.90 μm above the glass, respectively. The restriction of phages to the bottom layer of the *V. cholerae* biofilm most likely indicates that these phages initially attached to the underlying glass surface and were overgrown by the expanding *V. cholerae* cell group as it expanded from its initial position of attachment.
(PDF)

**S7 Fig. Within co-culture flow devices, *E. coli* cells exhibit similar levels *csgBAC* transcription independently of their distance from *V. cholerae*.** (**A**) *csgBAC* transcription as a function of distance of *E. coli* cells from the nearest *V. cholerae*. The data underlying this figure can be found in S1 Data.
(PDF)

**S8 Fig. *E. coli* and *V. cholerae* compete for space and nutrients with *E. coli* falling to a steady-state frequency of 2%–5% from a range of different starting frequencies.** (**A**) *E. coli* frequency in biofilm co-culture with *V. cholerae*, with and without the introduction of phages (*n* = 4). (**B**) *V. cholerae* absolute abundance in monoculture and in co-culture with *E. coli* after 120 h of biofilm growth (Mann–Whitney U test with *n* = 8, *n* = 16). The data underlying this figure can be found in S1 Data.
(PDF)

**S9 Fig.** *Vibrio cholerae* **biofilm architecture is maintained through time even as** *E. coli* **inclusions continue to grow and expand.** (**A**) Heatmaps of the merged neighborhood biovolume fraction for both cell types over the time course experiment shown in (**B**). (**B**) Time course of a dual culture biofilm of *V. cholerae* (purple) and *E. coli* (yellow), with phages being introduced into this system from 48 h onward. (**C**) Time course of a monoculture biofilm of *E. coli*, with phages introduced from 48 h onward. (**D**) Heatmaps of the merged neighborhood biovolume fraction for the time course shown in (**D**). The data underlying this figure can be found in S1 Data.
(PDF)

**S10 Fig. A diagram of the microfluidic device chamber design used for biofilm growth experiments in this study.** This example contains 4 parallel chambers, each with an inlet and an outlet port for connection to fluid inlet/outlet tubing. Technical replicate image stacks were taken from within the straight rectangular section between the rounded inlet and outlet ports of a chamber. The thinner, continuous channel surrounding the 4 separated chambers was connected to a wall vacuum line to apply negative pressure; this method discourages the introduction of air bubbles into the liquid-filled portions of the flow chambers.
(PDF)

## Acknowledgments

We are grateful to William Harcombe, Mary Lou Guerinot, and Alexandre Persat for feedback on earlier versions of this manuscript, and to members of the Nadell Lab and microbiology community at Dartmouth for feedback on the project. We are very grateful as well to Lanying Zeng for providing phage λD-mNeongreen cI857-mKate2 used for S3 Fig.

## Author Contributions

**Conceptualization:** James B. Winans, Benjamin R. Wucher, Carey D. Nadell.

**Data curation:** James B. Winans, Carey D. Nadell.

**Formal analysis:** James B. Winans, Carey D. Nadell.

**Funding acquisition:** Carey D. Nadell.

**Investigation:** James B. Winans, Carey D. Nadell.

**Methodology:** James B. Winans, Benjamin R. Wucher, Carey D. Nadell.

**Project administration:** Carey D. Nadell.

**Resources:** James B. Winans, Benjamin R. Wucher, Carey D. Nadell.

**Supervision:** Carey D. Nadell.

**Validation:** James B. Winans, Carey D. Nadell.

**Visualization:** James B. Winans, Carey D. Nadell.

**Writing – original draft:** James B. Winans, Carey D. Nadell.

**Writing – review & editing:** James B. Winans, Carey D. Nadell.

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
