## [Editor Report · Decision Letter 0]

31 Oct 2022

Dear Carey, 

Thank you for submitting your (revised, ex-eLife) manuscript entitled "Cell group architecture dictates phage exposure in multispecies biofilms" for consideration as a Research Article by PLOS Biology.

Your manuscript has now been evaluated by the PLOS Biology editorial staff, and I'm writing to let you know that we would like to consider your revised manuscript further (see my explanatory email sent separately).

However, before we can do so, we need you to complete your submission by providing the metadata that is required for full assessment. To this end, please login to Editorial Manager where you will find the paper in the 'Submissions Needing Revisions' folder on your homepage. Please click 'Revise Submission' from the Action Links and complete all additional questions in the submission questionnaire.

Once your full submission is complete, your paper will undergo a series of checks. After your manuscript has passed the checks it will be sent out for further assessment by the Academic Editor. To provide the metadata for your submission, please Login to Editorial Manager (https://www.editorialmanager.com/pbiology) within two working days, i.e. by Nov 02 2022 11:59PM.

Kind regards,

Roli

Roland Roberts, PhD

Senior Editor

PLOS Biology

rroberts@plos.org

---

## [Editor Report · Decision Letter 1]

2 Nov 2022

Dear Carey,

Thank you for your patience while we considered your revised manuscript "Cell group architecture dictates phage exposure in multispecies biofilms" for publication as a Research Article at PLOS Biology. This revised version of your manuscript has been evaluated by the PLOS Biology editors and the Academic Editor.

Based on our Academic Editor's assessment of your revision, we are likely to accept this manuscript for publication, provided you satisfactorily address the following data and other policy-related requests.

IMPORTANT: Please attend to the following:

a) Please change the title to something more explicit; we suggest something like "Cellular architecture of multispecies biofilms determines phage exposure" or even "Cellular architecture of multispecies biofilms determines each host species' exposure to phage" (I note that we found the phrase "Cell group architecture" somewhat opaque)

b) The Academic Editor was concerned that the graphs in Fig 4 didn't seem to change visibly during revision, despite the new data. To my eye, the graphs seem subtly different, and I see that the sample numbers have changed in the legend; however, the Academic Editor requested that you "either highlight the repetition and the subsequent changes to figure 4 or include a supplemental figure showing evidence of this repetition."

c) The Academic Editor also asked that you include a schematic or Figure of the initial microfluidic device used in this study as a supplementary Figure. ("With this growing field, I think it is important to properly document these devices as there is a lot of variation in their shape and design that impacts the reported results.")

d) Many thanks for your very thorough data provision. Please could you cite the location of the data clearly in all relevant main and supplementary Figure legends, e.g. “The data underlying this Figure can be found in S1 Data.”

We expect to receive your revised manuscript within two weeks. 

*Published Peer Review History*

*Press*

Sincerely,

Roli

Roland Roberts, PhD

Senior Editor,

rroberts@plos.org,

PLOS Biology

We require the original, uncropped and minimally adjusted images supporting all blot and gel results reported in an article's figures or Supporting Information files. We will require these files before a manuscript can be accepted so please prepare and upload them now. Please carefully read our guidelines for how to prepare and upload this data: https://journals.plos.org/plosbiology/s/figures#loc-blot-and-gel-reporting-requirements

DATA NOT SHOWN?

---

## [Editor Report · Decision Letter 2]

14 Nov 2022

Dear Carey,

Thank you for the submission of your revised Research Article "Multispecies biofilm architecture determines bacterial exposure to phages" for publication in PLOS Biology. On behalf of my colleagues and the Academic Editor, Jeremy Barr, I'm pleased to say that we can in principle accept your manuscript for publication, provided you address any remaining formatting and reporting issues. These will be detailed in an email you should receive within 2-3 business days from our colleagues in the journal operations team; no action is required from you until then. Please note that we will not be able to formally accept your manuscript and schedule it for publication until you have completed any requested changes.

Sincerely,

Roli

Senior Editor

PLOS Biology

rroberts@plos.org